# Long-Term Assessment of Antibody Response to COVID-19 Vaccination in People with Cystic Fibrosis and Solid Organ Transplantation

**DOI:** 10.3390/vaccines12010098

**Published:** 2024-01-18

**Authors:** Teresa Fuchs, Dorothea Appelt, Helmut Ellemunter

**Affiliations:** Department of Child and Adolescent Health, Paediatrics III, Cystic Fibrosis Centre Innsbruck, Medical University of Innsbruck, 6020 Innsbruck, Austria

**Keywords:** cystic fibrosis, COVID-19, SARS-CoV-2, vaccines, antibody response, transplantation

## Abstract

With the worldwide spread of SARS-CoV-2 disease, people with cystic fibrosis (CF), especially solid organ transplant recipients, have quickly been identified as a risk group for severe disease. Studies have shown low antibody response to SARS-CoV-2 vaccines in recipients of solid organ transplant compared to the healthy population. Information on immune response in CF patients with solid organ transplantation is limited, especially regarding long-term efficacy. The aim of this real-world study was a long-term assessment of humoral immune response induced by three and four doses of the SARS-CoV-2 mRNA vaccine. S1RBD and IgG antibodies were measured every 12 weeks over a period of 27 months in twelve CF patients (five liver and seven lung transplantation recipients). A total of 83.3% of our patients showed a positive antibody response after three doses of the SARS-CoV-2 mRNA vaccine. A sustained immune response was observed in both groups over the observation period, with liver transplant recipients showing higher levels than lung transplant recipients. This study is among the first to show long-term data with constantly elevated or even increasing antibody levels. We conclude that this effect is most likely associated with repeated boostering in terms of infections and booster vaccinations.

## 1. Introduction

Cystic fibrosis (CF) is an autosomal recessive metabolic disease that affects many organs. In addition to the involvement of the pancreas, intestines and liver, the lungs are particularly affected by the loss of function of the CF transmembrane conductance regulator protein [1]. A circle consisting of inflammation, mucus obstruction and infection drives the disease with the development of bronchiectasis as a long-term consequence [2]. Despite the therapeutic achievements in recent years, lung transplantation is still necessary as a final treatment option [3]. In the context of severe CF liver disease, liver transplantation is still a regularly performed procedure in patients with CF [4].

### Cystic Fibrosis and SARS-CoV-2

With the worldwide spread of SARS-CoV-2 disease in 2019, also called COVID-19, people with chronic respiratory diseases, including patients with CF, were included in the risk group for potentially severe disease progression [5]. Viruses such as influenza or respiratory syncytial virus can cause acute deterioration of lung function in patients with CF. Despite this known risk constellation, outcome data from patients with CF who developed COVID-19 soon showed better results than originally suspected [6]. It has been shown that a dysfunctional CFTR channel reduces viral entry and replication and may therefore protect against severe SARS-CoV-2 infection [7]. Data from the European Cystic Fibrosis Patient Registry showed that general symptoms were the most common in the context of a COVID-19 infection, including fever, fatigue, myalgia/arthralgia and headache, followed by increased cough and pulmonary exacerbation [8]. However, transplant status was still considered one of the main risk factors for serious outcomes and these patients were therefore given priority in the vaccination program [9]. In view of the rapid spread of the infection, vaccines were developed very quickly. Two anti-SARS-CoV-2 vaccines based on mRNA technology (Comirnaty^®^ (BNT162b2), Pfizer—BioNTech (New York, NY, USA) and Spikevax^®^ (mRNA-1273), Moderna—NIAID (London, UK)) have been approved [10,11]. Since people with solid organ transplantation require immunosuppressive treatment, their immune response to infections or vaccinations can be insufficient. Studies have shown that recipients of organ transplant have a poorer immune response to SARS-CoV-2 vaccinations than non-transplanted people [12]. Factors that were associated with poor antibody response included older age, deceased donor status, recent rituximab exposure and high doses of mycophenolate mofetil and/or immunosuppressive combination therapy [13].

We recently reported short-term antibody response of CF patients who had undergone liver (CF-LI) or lung (CF-LU) transplantation, following two or three COVID-19 vaccination doses [14]. We compared levels to recipients of solid organ transplants without CF as an underlying disease and could show higher rates of serological response in CF patients. We could also show that antibody levels in the CF-LI group were generally higher than in the CF-LU group, presumably due to the lower immunosuppression and lack of mycophenolate mofetil therapy [14]. Data on immune response to SARS-CoV-2 mRNA vaccination in CF patients with solid organ transplantation are limited, especially regarding long-term efficacy. Information on the antibody response to vaccinations and infections is important to further optimize the treatment of this patient group in view of the ongoing exposure to the virus.

The aim of this study was to assess humoral immune response over 27 months induced by three and four doses of SARS-CoV-2 mRNA vaccine.

## 2. Materials and Methods

As part of the routine outpatient check-ups carried out at least four times a year at the CF center Innsbruck, blood samples were taken to determine the antibody response starting in autumn of 2020. Antibodies against the receptor-binding domain of S1 subunit of spike protein (S1RBD) and antibodies against spike protein IgG were regularly measured. Abbot SARS-CoV-2 IgG II Quant Assay and ARCHITECT i system were used to detect S1RBD [15]. Analysis of SARS-CoV-2 IgG antibodies was performed using Liaison^®^ SARS-CoV-2 TrimericS IgG Assay [16]. The conversion factor from the original unit from instrument to validated unit was AU/mL × 2.6 = BAU/mL. Positive cut off for S1RBD was set at 7.1 BAU/mL for spike protein IgG at 33.8 BAU/mL [15,16]. Descriptive analysis was performed with IBM SPSS, version 29.0 and GraphPad Prism, Version 9.5.1.

## 3. Results

### 3.1. Patient Characteristics

Twelve patients with long-term data were included in in this real-world cohort study, of whom five received liver transplantation and seven received lung transplantation. In CF-LI, four patients were male and one was female, whereas in CF-LU, four were male and three were female. Patients were generally younger in the CF-LI group with a mean age of 29 years (range 22–48 years) compared to 44 years in CF-LU (range 33–63 years). Mean time interval since the performance of transplantation was 16 years in CF-LI (range 11–24 years) and 12.5 years in CF-LU (range 7–25 years).

Lung function showed abnormal values in CF-LI with 66% ppFEV_1_ (range 31–88%) compared to 80% (range 43–108%) in CF-LU according to global lung initiative [17]. Lung clearance index (LCI) measured by multiple breath washout is regularly measured in our center in all patients regardless of transplantation status. In line with lung function, there was an elevated LCI value in CF-LI compared to lung-transplanted patients (12 vs. 9). Body mass index (BMI) was comparable in both groups with 20 kg/m^2^ in CF-LU and 22 kg/m^2^ in CF-LI, respectively. In CF-LI, three out of five patients were treated with mycophenolate mofetil in combination with tacrolimus or everolimus. Two patients were treated with single-agent immunosuppression, either tacrolimus or cyclosporine, with one additionally receiving prophylactic azithromycin. None of the liver transplant recipients received cortisone therapy. In CF-LU, all seven patients received combination immunotherapy (five tacrolimus/mycophenolate mofetil and two tacrolimus/everolimus) and permanent treatment with cortisone. CF-LU patients had higher creatinine levels (1.8 mg/dL) compared to CF-LI patients (0.9 mg/dL), determined at the time of blood sampling for the first antibody measurement. Regarding chronic bacterial lung colonization, six patients in total were classified as chronically *pseudomonas aeruginosa* (PA)-infected according to modified Leeds criteria (three patients each in both groups). All patients with chronic PA infection and CF-LI were on long-term inhaled antibiotic therapy. Patient characteristics are shown in Table 1.

All patients were vaccinated at least three times with a SARS-CoV-2 mRNA vaccine (30 µg Comirnaty and/or 100 µg Spikevax) in 2021 and 2022, respectively. Regarding side effects and COVID-19 infections during this period, we refer to our primary study [14]. Only four patients (33.3%) received a fourth vaccination in winter 2022/2023 (30 µg Comirnaty).

### 3.2. SARS-CoV-2 Antibodies

Antibodies were measured every three months after receiving the third vaccination over an observation period of two years (mean 25, range 21–27 months) until September 2023. In CF-LI, a 100% response rate was seen after three vaccinations for S1RBD and IgG, respectively, which was sustained over two years. Three patients received a fourth vaccination and two suffered from COVID-19 infection during this period (one patient without receiving a fourth vaccination, while the other suffered a few weeks after the fourth booster vaccination). These patients experienced mild symptoms without the need for additional therapy or hospitalization. In CF-LU, 71.4% of patients reached a stable positive response rate after their third vaccination, in which IgG levels were elevated in all patients, but S1RBD values only in five. Two patients fell ill due to COVID-19 infection, again developing mainly mild symptoms (both of them did not receive fourth vaccination). These two patients showed relatively low antibody levels compared to the other lung transplant patients. One person was detected as COVID-19 positive without experiencing any symptoms. Only one patient received a fourth vaccination. Mean S1RBD antibody level two years after the first vaccination was 3481.6 BAU/mL (range 326.6–5721.9 BAU/mL) in CF-LI and 780.2 BAU/mL (range 3–2224.8 BAU/mL) in CF-LU. Mean IgG level was 1797.6 BAU/mL (range 668–2080 BAU/mL) in CF-LI and 1233.9 BAU/mL (range 16.4–2080 BAU/mL) in CF-LU. Changes in S1RBD and IgG antibodies are shown in Figure 1.

## 4. Discussion

As described in our previous study, overall, 83.6% of our patients had a detectable antibody response after three doses of the SARS-CoV-2 mRNA vaccine, where the response of CF-LI was clearly stronger than that of CF-LU [14]. Looking at both S1RBD and IgG trends in this study, there is an obvious decrease in the first months after the third vaccination in both groups, accompanied by an increase in CF-LI and a stabilization in lung-transplanted patients. The tendency of higher antibody values of CF-LI versus CF-LU continued over the entire observation period. Previously described factors influencing antibody levels that may also apply to our cohort are the younger age of CF-LI, leading to higher antibody levels, and the higher creatinine levels in CF-LU, leading to lower antibody levels [18]. However, these are only observations, as qualitative statistical calculations are not possible due to the small study cohort. Nevertheless, a persistent immune response is seen in both groups (Figure 1).

Data regarding the immune response to SARS-CoV-2 vaccination of patients with CF and organ transplantation are very scarce. A study by Lucca et al. investigated the antibody response of 18 lung-transplanted CF patients 24–28 weeks after the second vaccination [19]. They compared values to CF patients without a history of transplantation and were able to confirm existing data on antibody response in CF: while antibody levels are generally comparable to people without CF, they confirmed a significantly decreased immune response of the transplanted patients, especially in those undergoing mycophenolate mofetil therapy [19]. In their study, patients are younger with a mean age of 38.8 years compared to our cohort (44 years). Similarly, the time interval since the performance of transplantation is almost half as long compared to our data (6.5 vs. 12.5 years). Despite the presence of these known risk factors for poor antibody response in transplant recipients, our cohort still shows a better response [12]. Likewise, long-term data after three or four vaccinations are not available. 

In general, there is little information on long-term data regarding antibody levels after vaccination in people with solid organ transplantation. Toniuotto et al. investigated data on 143 liver-transplanted patients and compared results to healthy controls [20]. In COVID-19 naive patients, 78.8% had positive values for S1RBD 6 months after their second vaccination and none of the primary positively tested patients became negative within 6 months. Antibody response in COVID-19-recovered patients showed significantly higher levels than the naive group (100% response rate), although both had lower S1RBD values compared to healthy controls [20]. Recent data on 61 liver transplant recipients show IgG levels three months after the third SARS-CoV-2 mRNA vaccination in patients receiving everolimus immunosuppression. These data are comparable to our results although long-term data are also not addressed in this study [21].

A study by di Filippo et al. has shown impaired long-term immune response up to 9 months after the second COVID-19 vaccination in 119 healthcare workers caused by a lack of vitamin D [22]. It is known that due to pancreatic insufficiency and intestinal malabsorption, patients with CF are prone to vitamin deficiency [23]. Vitamin D deficiency is especially common in patients with CF and increases the risk of pulmonary exacerbations in children and adults.. Therefore, substitution therapy with vitamin D is recommended [23]. The levels of 25-hydroxy vitamin D are regularly measured in our patients and, if necessary, substituted until normal levels (>75 nmol/L) are reached. All our patients had normal 25-hydroxy vitamin D levels, so a link between the comparably high SARS-CoV-2 antibody levels and sufficient vitamin D levels in our cohort can be hypothesized.

Only four patients in our cohort received a fourth vaccination despite clear recommendations. Reasons for this hesitancy have yet to be investigated. An Italian working group conducted a questionnaire study in which fear of adverse events and perceived lack of efficacy were found to be the main reasons for hesitancy [24]. Although less than half of the patients received a fourth vaccination, half of them had a COVID-19 infection during the observation period. The two CF-LU patients suffering from mild infection had relatively low antibody levels compared to the other lung transplant patients despite having received three vaccinations. It can be argued that lung-transplanted patients with low antibody levels are associated with a higher risk of breakthrough infection. Taking all of this into account when looking at the long-term antibody values in our cohort, it is possible to maintain constantly elevated values over 27 months.

In conclusion, this study is one of the few that present antibody response in patients with CF and solid organ transplantation after SARS-CoV-2 vaccination. It is also among the first to show long-term data on immune response, using the same method during the observation period. Major limitations of the study are the small population size, which mostly only allows for descriptive statistics, the single-center study format, and the lack of a control group. We have shown constantly elevated or even increasing antibody levels in CF-LI and CF-LU over a period of two years after the first vaccination. This effect is most likely associated with a repeated boostering in terms of infections and booster vaccinations. Referring to the mild or subclinical symptoms in the course of a SARS-CoV-2 infection despite immunosuppression, we conclude that the vaccination was successful and therefore emphasize the importance of booster vaccination for this specific vulnerable patient population once more.

## Figures and Tables

**Figure 1 vaccines-12-00098-f001:**
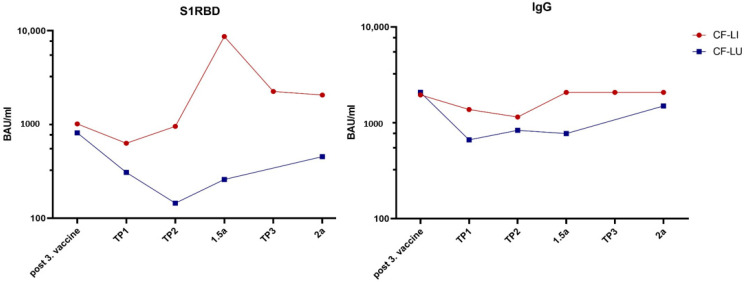
Comparison of mean S1RBD and IgG levels in CF-LI (in red circles) and CF-LU (in blue squares) over an observation period of 2 years showing stable antibody values. TP3 data in CF-LU are missing as only two patients appeared for this visit.

**Table 1 vaccines-12-00098-t001:** Characteristics of participants.

Item	Total	Liver	Lung
(N)	12	5	7
Sex n (%)			
Male	8 (67.0%)	4 (80.0%)	4 (57.2%)
Female	4 (33.0%)	1 (20.0%)	3 (42.8%)
Age (y) mean (range)	38 (22, 63)	29 (22, 48)	44 (33, 63)
Time since Tx (y) mean (range)	14 (7, 25)	16 (11, 24)	12.5 (7, 25)
FEV_1_% mean (range)		66.3 (31, 88)	80.0 (43, 108)
LCI median (range)		12.0 (8.8, 16.6)	9.0 (7.2,12)
(BMI/kg/m^2^) median (range)		20 (18, 21)	22 (17, 27)
CFRD n (%)	9 (75.0%)	4 (80.0%)	5 (71.0%)
Insulin therapy (%)	8 (67.0%)	3 (60.0%)	5 (71.0%)
Mycophenolate mofetil therapy n (%)	10 (83.0%)	3 (60.0%)	7 (100.0%)
Chron. PA colonization n (%)	6 (50.0%)	3 (60.0%)	3 (43.0%)
Creatinine (mg/dL) mean (range)	1.4 (0.6, 2.8)	0.9 (0.6, 1.4)	1.8 (1.3, 2.8)

## Data Availability

The data presented in this study are available on request from the corresponding author. The data are not publicly available due to privacy and ethical reasons.

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
