# Peer review of "Long-Term Assessment of Antibody Response to COVID-19 Vaccination in People with Cystic Fibrosis and Solid Organ Transplantation"

_vaccines, 2024, doi:10.3390/vaccines12010098_

Round 1

Reviewer 1 Report

Comments and Suggestions for Authors

This study delineates a case series of long-term antibody dynamics in cystic fibrosis. The Introduction and Discussion were well-written. However, the immunologic assessments and Results were unclear. Suggest adding more information to make it complete.

Major concerns.

1. Suggest adding the immunologic assay. Which tests do you use in this study? It is necessary to delineate the test and the original unit from the instrument.

I guess on the target and the cutoff.
Were there AdviseDx SARS-CoV-2 IgG II or SARS-CoV-2 IgG II Quant from Abbott and SARS-CoV-2 TrimericS IgG from Diasorin?

Please add information to the Methods following;
- The immunologic assay of both reagent and instrument
- Conversion factor from the original unit from instrument (AU/mL) to validated unit (BAU/mL).

For Abbott.

This instrument cutoff is 7.1 BAU/mL (50.0 AU/mL).
Suggest correcting the cutoff from 7.0 BAU/mL to 7.1 BAU/mL.

2. Line 109: "All patients were vaccinated at least 3 times with a SARS-CoV-2 mRNA vaccine (30μg Comirnaty and/or 50μg Spikevax)".

The Spikevax regimen is 100 μg in full-prime, and the recommended booster dose is 50 μg. However, some studies or in the immunocompromise may consider the booster use as 100 μg.

Please check the actual dosage in the Spikevax regimen in this study again.

3. Suggest creating the Table to delineate the antibody in all participants, not only two long-term follow-up participants in Figure 1. You can create a scatter or dot plot mimicking this table to make it more informative.

Comments.

1. Introduction. You may emphasise that solid organ transplant recipients require immunosuppressive treatment, leading to insufficient or unresponsive immune response.

2. "Pfizer-BioNTech® BNT162b2 and Moderna®mRNA-1273". The statement was wrong because Pfizer—BioNTech and Moderna (Moderna—NIAID) are manufacturer names, and ® meant registered trademark. You cannot use ® with the manufacturer's name.

Suggest using "(BNT162b2, Pfizer—BioNTech; mRNA-1273, Moderna—NIAID)".
Otherwise, you can use "(Comirnaty®, Pfizer—BioNTech; Spikevax®, Moderna—NIAID)".
OR "Comirnaty® (BNT162b2), Pfizer—BioNTech and Spikevax® (mRNA-1273), Moderna—NIAID".

3. Figure 1. To make it more informative, you may add some signs in the plot describing the period the patient got COVID-19 infection or second booster vaccination (4-dose).

Typos.

1. Line 45 suggests using COVID-19 instead of Covid-19.

2. Line 55, "anti-SARS- CoV-2", the gap between "-" and "CoV".

3. Lines 79, 115 "SARS-Cov-2".

Author Response

  1. Suggest adding the immunologic assay. Which tests do you use in this study? It is necessary to delineate the test and the original unit from the instrument. Please add information to the Methods following;

- The immunologic assay of both reagent and instrument
- Conversion factor from the original unit from instrument (AU/mL) to validated unit (BAU/mL).

For Abbott.

This instrument cutoff is 7.1 BAU/mL (50.0 AU/mL).
Suggest correcting the cutoff from 7.0 BAU/mL to 7.1 BAU/mL.

Response: Detailed information on the immunologic assays used in this cohort as well as conversion factor were added to the Methods. Cut-off for S1RBD was corrected from 7.0 to 7.1 BAU/ml as suggested.

  1. Line 109: "All patients were vaccinated at least 3 times with a SARS-CoV-2 mRNA vaccine (30μg Comirnaty and/or 50μg Spikevax)".

The Spikevax regimen is 100 μg in full-prime, and the recommended booster dose is 50 μg. However, some studies or in the immunocompromise may consider the booster use as 100 μg. Please check the actual dosage in the Spikevax regimen in this study again.

Response: After re-checking the patients' digital vaccination certificate and also consulting their GPs, it emerged that all patients had indeed been vaccinated with 100µg Spikevax . 
Many thanks for the information and the correction. 

  1. Suggest creating the Table to delineate the antibody in all participants, not only two long-term follow-up participants in Figure 1. You can create a scatter or dot plot mimicking this table to make it more informative.

Response: We originally tried to present antibody data in that way. However, due to the range of the data and long-term observation, the scatter plot showed a very unclear picture from which no qualitative information could be read.
Fig 1 does not only show two participants, it shows mean levels of antibodies of all of the patients. We therefore have decided that a linear graphic of the mean values over the period of 2 years is the best graphical representation.

Comments.

  1. Introduction. You may emphasise that solid organ transplant recipients require immunosuppressive treatment, leading to insufficient or unresponsive immune response.

Response: Thank you for very much for the hint, this has been added to the introduction (line 58).

  1. "Pfizer-BioNTech® BNT162b2 and Moderna®mRNA-1273". The statement was wrong because Pfizer—BioNTech and Moderna (Moderna—NIAID) are manufacturer names, and ® meant registered trademark. You cannot use ® with the manufacturer's name.

Suggest using "(BNT162b2, Pfizer—BioNTech; mRNA-1273, Moderna—NIAID)".
Otherwise, you can use "(Comirnaty®, Pfizer—BioNTech; Spikevax®, Moderna—NIAID)".
OR "Comirnaty® (BNT162b2), Pfizer—BioNTech and Spikevax® (mRNA-1273), Moderna—NIAID".

Response: This has been changed, thank you for the information (line 56).

  1. Figure 1. To make it more informative, you may add some signs in the plot describing the period the patient got COVID-19 infection or second booster vaccination (4-dose).

Response: Since this is real-life data in the context of clinical routine and not in a clinical study, all patients received their booster vaccinations at very different times. This in combination with time of infections in each affected patient, the graphic would become very busy and hard to read unfortunately.

Typos.

  1. Line 45 suggests using COVID-19 instead of Covid-19.
  2. Line 55, "anti-SARS- CoV-2", the gap between "-" and "CoV".
  3. Lines 79, 115 "SARS-Cov-2".

Response: Thank you very much for spotting and pointing out the typos, they have been corrected.

Reviewer 2 Report

Comments and Suggestions for Authors

The study investigates the long-term humoral immune response to SARS-CoV-2 mRNA vaccines in Cystic Fibrosis (CF) patients who have undergone solid organ transplantation, identified as a high-risk group for severe COVID-19.

Introduction:

Consider adding a sentence that more directly links CF complications to the increased risks posed by COVID-19.

You mention that "Data on immune response to SARS-Cov-2 mRNA vaccination in CF patients with solid organ transplantation is limited," but expanding on why this particular focus is crucial would be valuable.

The introduction briefly touches on the general response to SARS-CoV-2 vaccines in solid organ transplant recipients. A bit more detail on why these patients show a lower antibody response could provide a stronger rationale for the study.

Methods and results:

You mentioned that antibody response was measured in patients during routine visits. Were they consecutive patients, or was there a selection process?

The use of chemiluminescent microparticle immunoassay and the cutoff values for S1RBD and spike protein IgG are well-described. However, more information on the assay's sensitivity, specificity, and validation in this particular patient population would strengthen this section.

Discussion:

The discussion on the trends observed in antibody levels and potential factors influencing them (like age and creatinine levels) is insightful. However, be cautious about making definitive causal inferences without appropriate statistical analysis to support these claims.

Given the small sample size and the single-center nature of the study, how representative are these findings for the broader CF population with organ transplants?

Further exploration into why this hesitancy exists among your cohort, and how it might have influenced the study outcomes, would be valuable.

Author Response

Introduction:
Consider adding a sentence that more directly links CF complications to the increased risks posed by COVID-19.

Response: A sentence regarding this consideration was added (line 43).

You mention that "Data on immune response to SARS-Cov-2 mRNA vaccination in CF patients with solid organ transplantation is limited," but expanding on why this particular focus is crucial would be valuable.

Response: A statement regarding the importance of determining antibody values in this patient group was added (line 73).

The introduction briefly touches on the general response to SARS-CoV-2 vaccines in solid organ transplant recipients. A bit more detail on why these patients show a lower antibody response could provide a stronger rationale for the study.

Response: See line 62.

Methods and results:
You mentioned that antibody response was measured in patients during routine visits. Were they consecutive patients, or was there a selection process?

Response: There was no selection process. All patients who received a solid organ transplant at our centre were included in the study. The outpatient clinics are planned according to the bacterial status of the patients, so they were not consecutive.

The use of chemiluminescent microparticle immunoassay and the cutoff values for S1RBD and spike protein IgG are well-described. However, more information on the assay's sensitivity, specificity, and validation in this particular patient population would strengthen this section.

Response: More information has been added (s. methods).

Discussion:
The discussion on the trends observed in antibody levels and potential factors influencing them (like age and creatinine levels) is insightful. However, be cautious about making definitive causal inferences without appropriate statistical analysis to support these claims.

Response: You are very right and that is also the biggest limiting factor of the study. This was emphasized once again (line 156).

Given the small sample size and the single-center nature of the study, how representative are these findings for the broader CF population with organ transplants?

Response: Of course, this question always arises when the study population is very small. However, it should also be noted that CF is still a rare disease and fewer patients need to be transplanted. In addition, there are very few CF centres that have regularly measured Covid-19 antibodies in a routine clinical setting. And so far, there are hardly any publications on this topic. Our study can therefore provide a first insight into this area and can serve later meta-analyses.

Further exploration into why this hesitancy exists among your cohort, and how it might have influenced the study outcomes, would be valuable.

Response: This is correct, and this topic will be explored in more detail in a future paper. For the current publication, we do not yet have data to address or answer this question.

Reviewer 3 Report

Comments and Suggestions for Authors

Thank you for sharing your manuscript assessing the vaccine-related anti-COVID-19 immunoresponse among cystic fibrosis patients undergoing solid organ transplantation. Here some comments and suggested edits that could help to improve your article. 

L48: How do you define "general symptoms" in the context of your manuscript? 

L58: How was the healthy status of the comparative population confirmed? 

L73: Routine visits/Medical check-ups of CF patients? 

L74: What kind of patients com for routine visits at least 4-times per year?

L84: Which eligibility criteria did the 12 patients have to meet to be part of the study? How were the 12 patients selected? 

L120: At what time point during your research did the 2 participants suffer from COVID-19 infection in relation to the 4th vaccination?

L123: Are those the same 2 patients mention in L120? Please re-write L123-131 for more clarity as its content in relation to the participants included is not clear. 

Table 1: Please be consistent when reporting the participants' characteristics in terms of decimal places reported. 

Comments on the Quality of English Language

Please see above. 

Author Response

L48: How do you define "general symptoms" in the context of your manuscript? 

Response: “general Symptoms” in the European Cystic Fibrosis Patient Registry were defined as fever, fatigue, myalgia/arthralgia and headache. The corresponding sentence in the text has been clarified (line 50).

L58: How was the healthy status of the comparative population confirmed? 

Response: Unfortunately, authors in the cited publication who compared transplanted patients with healthy population do not define the healthy status any closer.

L73: Routine visits/Medical check-ups of CF patients? 

Response: According to the international guidelines for the treatment of cystic fibrosis patients, these patients are examined four times a year in outpatient clinics. Blood samples are taken as part of these visits routinely.

L74: What kind of patients com for routine visits at least 4-times per year?

Response:  Every CF patient treated at our centre regardless of age, sex, transplantation status and health status is coming at least 4 times per year for routine check-ups.

L84: Which eligibility criteria did the 12 patients have to meet to be part of the study? How were the 12 patients selected? 

Response: All transplanted patients who are treated in our centre and who have regular antibody levels were included in the study. Due to the already small number of cases and the fact that this is a real-world study, no further inclusion criteria were applied.

 L120: At what time point during your research did the 2 participants suffer from COVID-19 infection in relation to the 4th vaccination?

Response: In the CF-LI group 2 patients suffered from COVID-19. One of them never received fourth vaccination, the other had a positive test for COVID 19 few weeks after fourth vaccination. Both had mild symptoms (s. additional information, line 130).

L123: Are those the same 2 patients mention in L120? Please re-write L123-131 for more clarity as its content in relation to the participants included is not clear. 

Response: Line 128 to 133 addresses the CF-LI group.
Line 133 to 138 addresses the CF-LU group. Therefore, they are not the same patients as the different transplant groups are being looked at.

Table 1: Please be consistent when reporting the participants' characteristics in terms of decimal places reported. 

Response: Thank you, this has been revised and changed (see table 1)

Round 2

Reviewer 1 Report

Comments and Suggestions for Authors

Typos.

1. Line 125 "SARS-Cov-2 antibodies." Suggests using "SARS-CoV-2 antibodies."

Author Response

Line 125 "SARS-Cov-2 antibodies." Suggests using "SARS-CoV-2 antibodies.

Response: Thank you, line 125 has been corrected. 

Reviewer 3 Report

Comments and Suggestions for Authors

Thank you for sharing the revised manuscript that reads much better. 

Table 1: Please add "%" to the variable insulin therapy.

Comments on the Quality of English Language

Please see above. 

Author Response

Table 1: Please add "%" to the variable insulin therapy.

Response: Thank you very much, this was changed to the table.